# Use of Hu-PBL Mice to Study Pathogenesis of Human-Restricted Viruses

**DOI:** 10.3390/v15010228

**Published:** 2023-01-13

**Authors:** Jesús Emanuel Brunetti, Maksym Kitsera, César Muñoz-Fontela, Estefanía Rodríguez

**Affiliations:** 1Bernhard-Nocht Institute for Tropical Medicine, 20359 Hamburg, Germany; 2German Center for Infection Research, Partner Site Hamburg-Borstel-Lübeck, 38124 Braunschweig, Germany

**Keywords:** hu-PBL, SCID, NOD/SCID, NSG, DC, virus

## Abstract

Different humanized mouse models have been developed to study human diseases such as autoimmune illnesses, cancer and viral infections. These models are based on the use of immunodeficient mouse strains that are transplanted with human tissues or human immune cells. Among the latter, mice transplanted with hematopoietic stem cells have been widely used to study human infectious diseases. However, mouse models built upon the transplantation of donor-specific mature immune cells are still under development, especially in the field of viral infections. These models can retain the unique immune memory of the donor, making them suitable for the study of correlates of protection upon natural infection or vaccination. Here, we will review some of these models and how they have been applied to virology research. Moreover, the future applications and the potential of these models to design therapies against human viral infections are discussed.

## 1. Introduction

Animal models have been classically used to study the pathophysiology of human diseases. In particular, mouse models are frequently used since they are easy to handle, cheap to maintain, and there are numerous genetically modified strains and available reagents [1]. However, in the context of infectious diseases, mice cannot always be utilized because they either do not consistently recapitulate the disease observed in humans, or the pathogens studied do not cause infection in common laboratory mice. This is, for example, the case of many human-specific viruses, which can only infect human cells [2]. To address this issue, different ‘humanized’ mouse models have been developed, in which human cells or tissues are transplanted and studied in the context of infection. One common approach is the generation of mice with human immune systems (HIS) using recipient mice in which different aspects of the mouse immune system have been removed via genetic engineering [3,4]. The lack of mouse immune-competence allows human immune cells to engraft and develop into a fully or partially mature human immune system. Besides, mouse strains expressing human growth factors and cytokines [5,6], in addition to immune suppression, can be used to generate mice with specific immune cell subpopulations. In general, HIS mice can be generated by either the transplantation of human hematopoietic stem cells (HHSCs) or by the transplantation of mature peripheral blood mononuclear cells (PBMCs) [7]. The PBMC-based models, usually called hu-PBL (for human peripheral blood leukocytes), have the advantage that the generated HIS mice retain the immune memory of the donor [8,9]. These models offer an advantage for studies aimed to understand vaccine- or natural immunity-induced correlates of protection since they allow for the transplantation of memory cells from specific donors. 

## 2. Hu-PBL Mouse Models

### 2.1. Hu-PBL-SCID

The first human PBMCs (hu-PBL) transplantation in mice was described in SCID mice, an immunodeficient version of the C.B17 mouse strain [8]. The SCID (for Severe Combined Immune Deficiency) mouse strain, characterized for the first time in 1983 [10,11], carries an autosomal recessive mutation in the *Prkdc^scid^* allele of chromosome 16. Therefore, it has a defective recombination mechanism that affects the normal development of T- and B-cell receptors (TCR and BCR, respectively), rendering these cells functionally immature [4,12,13]. Since this first report, many studies have demonstrated that, among other things, the number of cells transplanted and the injection route are crucial for successful PBMC engraftment, the intraperitoneal injection being the most effective route [8,13].

Although transplanted PBMCs also contain hematopoietic stem cells (HSC), the quantity and quality of these cells makes them unable to recolonize mouse lymphoid tissues, remaining in the peritoneal cavity without being detected in the spleen, Peyer’s plaques, thymus, or other organs such as the liver [9,14].

Upon transplantation, human antigen-presenting cells (APCs), T helper cells, and B-lymphocytes engraft and may survive for at least 20 weeks in recipient SCID mice. Human-specific immunoglobulins are also detected in the model, with human IgG levels peaking 60 days upon engraftment [12]. On the contrary, IgM peaked at 30 days and then gradually decreased.

Regarding B cells, their engraftment is rare, probably due to the low presence of helper T cells. Finally, it has been shown that transplanted dendritic cells (DCs) retain a solid migratory capacity towards lymphoid tissues, and even without stimuli they can be found in the lungs, the thymus and the spleen [15]. However, the process by which these cells mature in this mouse model is not yet totally understood.

With respect to T cells, they appear in the blood 2 to 3 weeks after transplantation and can be detected in the spleen-active human CD3^+^ cells, having CD4^+^ or CD8^+^ phenotypes and expressing HLA-DR, which means they are activated [9,16]. After four weeks, CD3^+^ cells proliferate in the liver and the lungs; in the spleen, CD8^+^ and CD4^+^ cells can be found in the red and the white pulp, respectively [9,16]. Most of these cells express CD45R0 [9,17], which suggests that they are mature, and they are HLA-DR^+^ Ki67^+^, indicating that they are active. CD3^+^ cells can also be found later in the thymus and the bone marrow, primarily CD8^+^ cells with an activated phenotype (HLA-DR^+^ Ki67^+^) [16]. Despite the expression of activation markers, it should be emphasized that memory T cells can become anergic due to insufficient activation via the mouse APCs [9,17].

Since certain NK cells and other immune system components are still present in these mice, mild to moderate Graft-vs-Host-Disease (GvHD) development is frequently observed [8,12]. Numerous studies employing anti-NK cell antibodies [13,18,19,20,21], irradiation [22], hormones [23,24,25], interleukins or chemokines [26,27,28] have improved engraftment outcomes and decreased the appearance of GvHD.

### 2.2. Hu-PBL-NOD/SCID

The second major shift in the evolution of humanized mouse models was the establishment of NOD/SCID mice by backcrossing SCID and NOD (Non-Obese Diabetic) mice [3,4]. These animals overcame some of the problems that SCID mice still presented, such as the presence of mouse monocytes, granulocytes and NK cells, which reduced engraftment and resulted in the development of B-cell lymphomas in some cases [4,5,29,30]. The NOD background in these mice results in a partial impairment of the function of NK cells and macrophages, in a decrease in cytokine production and lack of mouse complement system activation [3,31,32,33,34]. Although these modifications improve engraftment, the number of T lymphocytes that can be found in peripheral blood after transplantation is still low. To overcome this problem, some studies have shown that the administration of IL-15 after irradiation and PBMC transplantation in these mice improves the number of T cells engrafted in primary and secondary lymphoid organs [35]. The fact that sublethal irradiation doses are still required for successful engraftment remains a limitation to the use of this model, since the irradiation may harm the blood vessels and cause the engrafted cells to perish as a result of prolonged vascularization [36].

Like the hu-PBL-SCID model, hu-PBL-SCID/NOD mice show a T cell-activated phenotype in the first-week post-transplantation, which becomes an anergic phenotype in later weeks. In addition, overall engraftment depends on the presence of CD4^+^ cells and the interaction between human PBMCs and host NK cells [37].

### 2.3. Hu-PBL-NSG

The introduction of the *IL-2Rγ* gene deletion in the NOD/SCID background, resulting in the NSG mouse strain, was a big step in developing hu-PBL models [3,4,31,32,33,34]. This new mouse strain allows for higher human engraftment because it is radiation sensitive, B and T cell deficient, and has defective NK cells due to the lack of IL-15 signaling and CD11c^+^ DCs, which reduces IFN production. This model shows a high percentage of human PBMCs 4 weeks after transplantation and the survival of T cells beyond ten weeks [38]. Human immune cells are detected in the liver, lymph nodes, spleen and other secondary tissues; however, CD19^+^ cells are mainly located in the lymph nodes and spleen. Within the T cell population, CD4^+^ and CD8^+^ cells have different distributions in the spleen compared to the hu-PBL-SCID model, with CD4^+^ occupying the red pulp and CD8^+^ the white pulp.

Similar to previous models, hu-PBL-NSG mice show signs of GvHD 16 weeks after transplantation [38]. The presence of CD4^+^ naive T cells (Tn) upon transplantation seems to trigger the development of GvHD [39]. On the other hand, CD4^+^ memory T cells (Tm) seem to function as APCs and produce IL-21, which improves B-cell maturation in this model [39,40].

### 2.4. Hu-PBL/DC

Two variations of the hu-NSG model have been developed to improve the presence of monocyte-derived dendritic cells (moDCs) after transplantation. The first model was initially established to assess the effectiveness of cancer DC-based vaccines [41]. In these studies, donor monocytes (PBMC CD14^+^ fraction) were cultured and matured ex vivo to obtain moDCs through incubation with IL-4 and GM-CSF, and different compound cocktails. Then, they were transfected with the antigen of interest and then transferred to mice that were previously transplanted with total PBMCs. These experiments showed that different maturation cocktails had different impacts on vaccine protection without affecting the levels of engraftment [41]. In the second approach, CD14+ monocytes were derived to moDCs in culture with IL-4 and GM-CSF [42]. Five days later, they were transduced with an adenovirus-expressing GFP and matured with IFN-γ and CD40L. Then, they were injected in mice previously transplanted with purified T and B cells. These models have been widely used in cancer research; however, in the field of viral infections, its use has been limited to one study on Ebola virus (EBOV) pathogenesis [43].

## 3. Research in Virus Immunology Using hu-PBL Mouse Models

Hu-PBL mice have been used to study the immune response to different human-specific viral infections (Figure 1). They have also served to study correlates of immune protection and vaccine efficacy. Some examples are reviewed here below.

### 3.1. DNA Viruses

Epstein Barr virus (EBV). EBV belongs to the *Herpesviridae* family of DNA tumor viruses. It has a double-stranded DNA genome, which encodes 80 different proteins. EBV has been associated with several cancers and it is the etiological agent of infectious mononucleosis [84]. EBV mainly infects epithelial cells and B cells, and it can respectively cause acute or latent infection in these cell types, similar to other herpesviruses.

The first report in which PBMCs were engrafted in SCID mice also showed the development of B-cell lymphomas due to the transplantation of EBV-positive donor cells, which caused the transformation of the transplanted B cells into lymphoma cells [8,44]. Consequently, EBV became the first viral infection studied in hu-PBL-SCID mice. Since then, hu-PBL-SCID mice have been used to understand EBV biology, immune response and transformation capacity. Different studies showed that human growth hormone therapy, antibodies that block NK cells or activated T cells had a negative impact on engraftment in these mice, while they seem to increase the risk of lymphoma development [22,24,45]. For instance, the daily treatment of hu-PBL mice engrafted with EBV-positive PBMCs with recombinant IL-2 reduced the development of EBV-related tumors [46]. This study also showed that the NK and T CD8^+^ cells were involved in the lymphomagenesis inhibition, since the in vivo depletion of NK-cells with an antibody or the engraftment of CD8-depleted PBMCs resulted in the appearance of lymphomas in 100% and 75% of the engrafted mice, respectively. Conversely, IL-10 promoted the growth of tumor B cells through an autocrine loop. This has been shown in hu-PBL mice, in which EBV positive lymphomas were induced. These animals had higher blood levels of IL-10 compared to those mice that did not develop tumors [47]. This increase in IL-10 was probably due to the presence of EBV in those lymphoma cells, since it is known that EBV infection can increase IL-10 production in B cells and that this has been related to cell death inhibition and, therefore, to tumor growth [48].

Anti-CD40 or anti-CD20 therapy has also been demonstrated to reduce the risk of malignant lymphomas induced by EBV. In a report, anti-CD40 antibodies were administered to SCID mice that had been depleted on NK cells and engrafted with several EBV lymphoma cell lines, which express CD40 as a surface marker [49]. This treatment increased survival rate and impaired tumor growth. In line with these results, another report showed that the administration of anti-CD40 or anti-CD20 antibodies to hu-PBL-SCID mice engrafted with EBV-positive PBMCs diminished the lymphomagenesis [50]. Interestingly, since these antibodies have diverse effects on B-cell engraftment, the mechanisms by which they impair the growth of B-cell tumors are different. When the surviving mice were examined for the presence of human B cells, it was shown that animals treated with anti-CD40 had a larger percentage of B-cell engraftment than mice treated with a control antibody, but mice treated with anti-CD20 had cleared B cells [50].

Contrary to what has been observed in SCID mice, the incidence of EBV-induced lymphomas is lower in NOD/SCID mice after 16 weeks of transplantation with EBV-positive PBMCs. Furthermore, NOD/SCID mice do not exhibit any symptoms of lymphoma development at four weeks [37], different to what has been observed in the SCID model [51]. Reducing the number of CD8^+^ cells by injecting an anti-CD8 antibody increased the risk of developing lymphoma in NOD/SCID transplanted mice. Accordingly, the administration of purified CD8^+^ from the same donor restored the levels of CD8^+^ cells and lowered the risk of lymphoma development. Interestingly, no effect was observed when CD4^+^ cells were administered to CD8-depleted or NK cell-depleted hu-PBL NOD/SCID animals. These results indicated that the development of B-cell lymphomas after engraftment is regulated by CD8^+^ cells.

Hepatitis B virus (HBV). HBV, a virus from the *Hepadnaviridae* family, is a partially double-stranded DNA virus that infects liver cells. HBV infection still poses a global health problem worldwide. Despite the availability of a vaccine, many individuals still develop chronic HBV infection and, therefore, have the potential to develop fatty liver, cirrhosis and hepatocellular carcinoma [85]. Hu-PBL-NOD/SCID mice have been used in vaccination studies against HBV. Mice reconstituted with healthy PBMCs mixed with yeast-produced antigenic HBV proteins (i.e., rHBcA or rHBeAg) have been used to analyze antibody production. On the one hand, anti-HBcAg IgM levels increased seven days following vaccination and were detectable in some animals after 15 days [52], while IgM production against HBeAg could not be detected. Surprisingly, these mice did not show any IgG seroconversion. On the other hand, IgG and IgM production was detected in the blood when mice were reconstituted with PBMCs from persistently HBV-infected or convalescent donors, even without administering the antigen to the mice [52].

### 3.2. Retroviruses

Human immunodeficiency virus (HIV). HIV is the agent of acquired immunodeficiency syndrome (AIDS), the final and severe phase of the infection. HIV is a retrovirus, meaning its RNA genome can integrate into the host genome upon retrotranscription into DNA [86]. Forty years after its discovery, HIV has now spread worldwide, and although there are many drug-based therapeutic options to maintain a constant negative viral level, no vaccine has yet been approved, and this virus still poses a threat in underdeveloped countries where those therapies are unavailable. The patient’s clinical symptoms in the early reports of AIDS caught the scientific community’s attention. Because of this, HIV was among the first viruses to be investigated in SCID mice [53]. These studies showed that mice receiving PBMC transplants from healthy donors were vulnerable to HIV infection, either by the virus itself or by lymphoblastoid T cells that had already been infected with HIV [53]. Additionally, following infection, only CD8^+^ CD45R0^+^ cells were recovered, and it was demonstrated that the loss rate of CD4^+^ lymphocytes depended on the inoculated strain [54,55]. These results were corroborated in later studies [56] which also showed that T cells displayed HLA-DR and CD25 markers, suggesting that they might be activated. The production of several proinflammatory Th0/Th2 cytokines concurred with this activation. In addition, the percentage of depleted CD4^+^ cells was influenced by the activation status of the transplanted cells and by the exposure to T-cell tropic or Macrophage-tropic viruses [57]. The memory T-cells phenotype and CCR5 expression were linked to these results.

In another study, a hu-PBL-SCID mouse model with widespread and permanent HIV infection was developed to reduce the anergic state of T cells after engraftment. Repeated injections of CD4^+^ T cells during four weeks after infection and transduction with IFN-β decreased HIV viral load, suggesting a potential infection eradication [58]. In line with this, one research has proposed the SCID models as the first avatar model reconstituted with PBMC from HIV-positive donors [59], showing similar results to those obtained when mice were infected after transplantation.

Hu-PBL-SCID mice have also been utilized to examine the effectiveness of anti-HIV drugs and antibodies against HIV infection. It was shown, for example, that the administration of monoclonal antibodies targeted against the binding site between the viral protein gp120 and the CD4 receptor reduces viral entry [60]. However, broad-spectrum antibody therapy does not have the anticipated impact on the clearance of infection [61,62]. Regarding antiviral drugs, rapamycin treatment in infected hu-PLB-SCID mice increased the ratio of CD4^+^ lymphocytes and decreased the levels of proviral integration and viral RNA in the blood [48]. Further research showed that treating hu-PBL-SCID mice with IgG2 inhibits HIV and protects these animals from infection [63].

Finally, the hu-PBL-SCID mouse model has been used to evaluate the efficacy of dendritic cell (DC)-based vaccines [64,65,66,67]. According to several studies, intrasplenic immunization of PMBC-engrafted SCID mice with HIV-inactivated primed DCs led to the development of antibodies against gp41, initially IgM and later IgG [64,65]. Furthermore, the antibody response was improved if DC development was induced via treatment with IFN/GM-CSF rather than with IL-4/GM-CSF [50]. Additional studies showed that DC-vaccinated mice were resistant to HIV challenge and that this immunization also induced the development of CD8^+^ lymphocytes specific against gag and pol HIV proteins [65,66].

When NOD/SCID mice emerged, new mouse models were established to study HIV infection and vaccine protection. Indeed, the first studies showed that the NOD/SCID model was better than the SCID model [68]. Splenocytes from naive PBMCs-transplanted NOD/SCID and SCID mice and then HIV-infected were compared. Ex vivo co-culture of these splenocytes with fresh PBMCs showed a higher p24 antigen production when splenocytes came from NOD/SCID mice. Moreover, 80% of the NOD/SCID co-cultures were HIV PCR positive, even if they were p24-negative.

Regarding vaccine testing against HIV, a herpes simplex virus (HSV)-based approach was tested in hu-PBL-NOD/SCID mice [69]. DCs from a naïve donor were infected with HSV expressing HIV gp120 protein (HSVgp120) or the β-galactosidase enzyme (HSVLac) as a control. Mice transplanted with HSVgp120-infected DCs produced significantly higher HIV-gp120-specific IgG levels than those that received the control HSVLac DCs. Moreover, IgG levels did not change with the second round of DC immunization. In addition, when mice were immunized with HSVgp120 DCs and challenged with HIV R5-strain, they had less expression of HIV p24 protein in the blood compared to those who received the HSVLac-DCs, suggesting that the HSV-gp120-vaccinated mice were partially resistant to HIV infection.

Finally, the hu-PBL-NSG model was proposed as a better model for studying HIV infection [38]. In these mice, HIV induced a decrease in the CD4^+^ T cells two weeks after infection [38,70], which correlated with an increase in p24 levels in plasma. Interestingly, the kinetics of CD4^+^ cell depletion in these mice depended on the virus strain used, similar to what was observed in the hu-PBL-SCID model [70]. Consistent with these findings about p24 levels, one study investigated the presence of p24^+^ cells in the meninges and cortex of hu-PBL-NSG mice infected with two distinct HIV strains using an immunofluorescence assay. Here, p24 colocalized with the CD3 marker, suggesting that CD4^+^ cells might be responsible for the viral spreading to the neurological system [70]. Besides, in the mouse cerebral cortex, the transcription levels of Iba-1 and GFAP, markers of microglia and astrocyte activation, respectively, correlated to p24 levels. These results indicated that the neurological damage observed in mice might be strongly associated with the cerebral expression of p24.

Other studies in hu-PBL-NSG mice have focused on studying the dynamics of HIV reservoirs in people on antiretroviral treatment (ART) [71,72]. CD4^+^ cells from ART-treated HIV-negative patients were transplanted into NSG mice. Eight weeks after, the number of CD4^+^ cells decreased while the viral load in the blood increased. This would suggest that proviruses can be reactivated following T cell activation and proliferation, which would model what occurs after discontinuation of ART therapy in humans.

Hu-PBL-NSG mice have also been used to show how HIV infection can potentially cause colon cancer development by altering β-catenin expression [73]. HIV-infected or uninfected mice were injected with a colon cancer cell line. After two weeks, HIV-infected mice developed more extensive tumors than uninfected ones. In addition, an immunohistochemical (IHC) analysis of tumor tissue showed that HIV-infected mice had higher expression of β-catenin. Nonetheless, the treatment with nelfinavir, zidovudine, and efavirenz (antiretroviral drugs) decreased the size of these tumors and the expression of β-catenin, suggesting that the virus might provide a microenvironment that encourages tumor cell development.

The antiviral efficacy of various drugs, such as zidovudine, indinavir and atazanavir, has also been assessed in hu-PBL-NSG mice [38]. Five weeks after receiving this medication, infected hu-PBL-NSG mice showed a reduction in viral load in plasma; however, once the treatment was stopped, p24 protein reappeared in blood, which would also mimic the possible effects of ART dropout.

The effectiveness of siRNA delivery against HIV was assessed in hu-PBL-NSG mice using a CD7-based antibody and an arginine oligopeptide [74]. One week following reconstitution with PBMCs from HIV-positive donors, mice were given two sequential injections of antibody–siRNA complexes. Mice receiving siCD4 showed a considerable and gradual reduction in the expression of this receptor in transplanted cells. Additionally, a weekly injection with a particular siRNA against CCR5, Vif, or Tat proteins in HIV-infected mice rescued the initial fall in CD4 cells caused by the viral infection. Furthermore, antiviral siRNAs dramatically reduced viral loads in all treated mice. In another study [75], hu-PBL-NSG mice were reconstituted with PBMCs from HIV-positive patients (ART-treated or untreated). Before transplantation, these cells were transduced with different HIV-based lentiviruses. Mice transplanted with cells that have been transduced with Tat shRNA-miR lentivirus showed an increase in the CD4^+^ cell proportion that decreased after 43 days. Conversely, mice transplanted with cells transduced with a pool of seven shRNA-miRs showed a constant increase in the CD4^+^ cell fraction even beyond 43 days after transplantation.

Previous reports have shown that neutralizing antibodies in HIV-infected hu-PBL-NSG mice impaired the reduction of the CD4^+^ cell population [38]. Recently, engineered antibodies have been tested in this mouse model [76]. For example, LSEVh-LS-F is a designed hexavalent protein that carries a gp120-binding domain and a CD4-binding domain. To test its ability to neutralize HIV, hu-PBL-NSG mice were infected with two strains of HIV expressing a reporter gene, one of which was susceptible to the VRC01 antibody (a human IgG1 monoclonal antibody that neutralizes a wide range of HIV strains), while the other was resistant. Upon infection, administration of VRC01 only prevented the replication of the VRC01-sensitive strain, while LSEVh-LS-F strongly suppressed the replication of both viral strains. Additional experiments removing NK cells from PBMCs, prior to transplantation in NSG mice, indicated that these cells mediate the effect of the antibody mentioned above.

The efficacy of an antibody against CCR5, Maraviroc (MVC), was tested in hu-PBL-NSG mice [77]. In these studies, mice infected with HIV were treated or not with Maraviroc at 12 weeks post-infection. Treatment was performed twice a day for three weeks. During the first week of treatment, both mice had a similar proportion of CD4^+^ cells. By the second week, untreated infected animals displayed fewer CD4^+^ cells than treated infected animals. MVC therapy also gradually reduced the percentage of CD8^+^ cells in infected mice. Nevertheless, the most exciting finding was that MVC treatment decreased the number of human HLA-DR^+^ cells, the viral protein expression, and the viral load in the brain of these animals. Moreover, treatment with Maraviroc boosted the expression of Claudin-5, ZO-1, and ZO-2, tight junction proteins in endothelial cells. Besides, the MVC treatment reduced the presence of amyloid in the brain and increased it in the plasma. In summary, these results suggest Maraviroc as a potential treatment against the consequences of HIV infection in the nervous system.

Anti-HIV treatment involving CAR T cells has also been explored in hu-PBL-NSG mice. These cells produce a genetically modified chimeric CD4 antigen receptor (CAR) that fuses the cellular activating domains of the TCR to an extracellular antigen-binding domain, allowing T cells to recognize antigens whether they have been processed or not [78]. Since T cell engraftment in NSG mice results in GVHD, these animals are appropriate for investigating this type of therapy in the context of HIV infection and GvHD-related symptoms. As the amount of CD4 cells decreases due to the infection, GvHD becomes milder, and mice are healthier. NSG mice transplanted with CD8^+^ T cells expressing different CARs and subsequently infected with HIV had larger numbers of CD4^+^ cells and undetectable viral loads, indicating that the CD8^+^ cells preserved the CD4^+^ cells [78]. In a different experiment, in mice harboring a latent HIV infection, the effect of CD8^+^ CAR T cells treatment was studied in the context of ART treatment interruption and rebound. Mice that had undergone treatment with CD8^+^ CAR T cells expressing a co-stimulatory domain (i.e., 4-1BB, CD28) had an increase in the number of CD4^+^ cells, while mice that received CAR T cells without any of these domains had a reduced number of CD4^+^ cells. Another study used HIV-infected PBMCs that were first treated with generated same-donor CAR T cells before transplantation into NSG mice. This approach also diminished the infection in these mice without changes in the proportion of CD8^+^ and CD4^+^ cells [79].

Other alternative therapies have been assayed in this mouse model. For example, treating HIV-infected hu-PBL-NSG mice with an IL-15 superagonist lowered viral gene expression levels while increasing NK cell number and activation, and CD4^+^ and CD8^+^ concentration in the spleen of these animals [80]. Furthermore, NK cell depletion before engraftment negatively influenced the efficacy of the IL-15 superagonist against HIV infection. These data suggest that NK cells mediated virus eradication in this in vivo model.

Lastly, other studies have assessed the role of IFN and IFN-stimulated genes (ISG) against HIV infection employing hu-PBL-NSG mice [81]. hu-PBL-NSG mice were treated before and after HIV infection with different doses of pegylated IFN (IFN bound to polyethylene glycol, PEG). In these mice, the decrease of CD4^+^ cells depended on the IFN concentration assayed, compared to untreated mice. In another set of additional experiments, hu-PBL-NSG mice infused with plasmids expressing different IFN-I subtypes showed a booster in ISG expression, which was maintained over time. Upon infection with HIV, these animals did not show a decrease in the number of CD4^+^ cells, while mice infused with a plasmid control had a decrease in the CD4^+^ cell number upon HIV infection. These results highlighted that only certain IFNs elicited a complete response later in the infection.

### 3.3. Single-Stranded Positive-Sense RNA Viruses

Dengue virus (DENV). Dengue virus is an arbovirus transmitted by different mosquitoes of the *Aedes* species. It belongs to the *Flaviviridae* family, and it causes mild to severe diseases, such as dengue fever, dengue hemorrhagic fever and dengue shock syndrome. The viral genome consists of a single-strand positive sense RNA with a unique open reading frame (ORF) that encodes a polyprotein cleaved into different mature proteins inside the infected cells [87]. It has been shown that, in hu-PBL-SCID mice generated by transplantation of donor PBMCs and infected with DENV-1, virus replication did not take place in most of the cases. Among all samples collected at different time points after transplantation, DENV-1 was only isolated in one of them. However, when mice were transplanted with DENV-1-infected monocytes and autologous PBMCs, no mice produced DENV-1-specific antibodies and from two of eight mice DENV-1 was isolated [82]. Other experiments have shown that antibodies against DENV-2 can be produced in this mouse model up to 7 weeks after transplantation with PBMCs from a DENV-2-convalescent donor and showed efficacy even against DENV-1 [82]. The reduced number of mice that become virus positive upon infection, and therefore, the low percentage of mice that show clinical signs, could be due to a poor lymphocyte infection or to a heterogeneous cell engraftment.

Severe acute respiratory syndrome coronavirus 1 (SARS-CoV-1). SARS-CoV-1 belongs to the *Coronaviridae* family. Its genome consists of one molecule of RNA, with a Poli-A tail and many ORFs, implying that several subgenomic RNA messengers are generated during its replication. Each subgenomic RNA encodes for a viral protein. In 2003, an epidemic of SARS-CoV-1 started in China and extended to at least 30 countries, causing mild to severe disease [88]. The hu-PBL-NSG model has been developed to test DNA-based vaccines against SARS-CoV-1 [83]. Upon transplantation of PBMCs from healthy donors, hu-PBL-NSG mice were injected with plasmids encoding SARS-CoV-1 M, N, or S proteins. M and N plasmids induced cytotoxic T cell activation and proliferation in hu-PBL-NSG mice. Interestingly, the production of neutralizing antibodies was observed only in mice immunized with the SARS-CoV-1 N or S plasmids.

### 3.4. Single-Stranded Negative-Sense RNA Viruses

Ebola virus (EBOV). Ebola virus (EBOV) belongs to the Filoviridae family and has caused different outbreaks with significant case-fatality rates in Africa. In 2014, EBOV caused an unprecedented epidemic in West Africa, raising worldwide public health concerns [89,90,91,92,93,94].

The hu-PBL/DC model has been recently used to study the immune response to Ebola virus infection and its role in Ebola virus disease (EVD) pathogenesis [43]. By transplanting CD14-depleted PBMCs and ex vivo EBOV-infected moDCs in NSG-A2 mice (a variant of the NSG model, which expresses the human leukocyte antigen A2 or HLA-A2), the authors showed that the human leukocyte antigen (HLA)-T cell receptor (TCR) crosstalk between T cells and infected DCs marked the outcome of infection in hu-PBL mice. Thus, the transplantation of HLA-TCR-matched T cells and EBOV-infected DCs from naïve individuals resulted in lethality in the model, whereas HLA-TCR mismatch resulted in survival. Furthermore, this model served to characterize the specific contribution of T cells and antibodies for protection. When hu-PBL mice were generated via the transplantation of EVD survivor T cells and antibodies, both components of the host immune response were needed to protect mice from re-infection. Antibodies or T cells alone were not protective. These results highlight the importance of both the humoral and the cellular immune responses during EBOV infection and suggest that the interaction between T cells and DCs is crucial for EVD development. Interestingly, this seems to be a rather specific feature of the Ebolavirus genus since parallel experiments with the Lassa virus (LASV), a hemorrhagic fever virus from the *Arenaviridae* family, did not show any severe pathogenesis or disease development when mice were transplanted with CD14-depleted PBMCs and LASV-infected moDCs. These results suggest that other immune compartments might play a role in LASV pathogenesis apart from T cells and moDCs [95,96,97]. Avatar-humanized mice seem to be an excellent model for studying the immune pathogenesis of Filoviruses.

## 4. Conclusions and Perspectives

Different humanized mouse models have been used to better understand human-specific viral infections. In general, these models offer the opportunity to study viral pathogenesis and treatment options in a human-like immune environment. Since this model conserves the immune memory of the donor, it may allow future studies on correlates of protection, original antigenic sin, virus reactivation and persistence and other processes in which immune memory plays a central role. Despite their potential, the exploration and development of these models in the context of infection is in its infancy [98,99,100].

Although in this platform, the immune system is not completely reconstituted, as in stem cell transplantation (HSC)-based models, the main disadvantage is that hu-PBL models use already mature immune cells for their generation, making them faster in developing GvHD. Compared to HSC models, the symptoms of GvHD appears earlier. Nonetheless, different approaches have been implemented to decrease GvHD in hu-PBL mice including the administration of interleukins, chemokines or hormones after irradiation. Recently, a new hu-PBL model, hu-PBL-TKO (Triple KnockOut) animals, which are defective for Rag1, IL-2Rγ and CD47 genes and have only been utilized for HIV research, reproduced similar disease features as to the ones observed in hu-PBL-NSG mice [101]. Interestingly, hu-PBL-TKO mice have shown a delayed onset of GvHD upon human immune cells transplantation compared to the H-PBL-NSG mice, which makes them attractive for the investigation of chronic viral infections, such as HIV or HBV, along with the hu-PBL-NSG transplanted with memory CD4^+^ cells [39,102].

The virus’s unique tropism for human receptors absent in mice is another downside of the hu-PBL model. Such is the situation with SARS-CoV-2, which has the membrane protein hACE2 as its main receptor [103]. Transgenic mice that express hACE2 transiently or constitutively have been established to address this problem [104,105,106,107]; however, they do not allow the engraftment of human cells, since they were made in a wild-type background [105,108,109,110]. In order to overcome this caveat, Jackson Laboratories (https://www.jax.org/ (accessed on 5 January 2023)) have recently developed different NSG mice that express the hACE2 receptor (strains 034901, 035959 and 035002) [111,112,113]. This opens new opportunities for studying SARS-CoV-2 pathogenesis in a human immune system environment. The integration of similar transgenic platforms and hu-PBL models will therefore enable the investigation of, for example, correlates of protection in convalescent and immunized patients against different viruses.

However, the transgenic expression of human receptors is not valid for other tissue-specific viruses. For instance, to explore the pathophysiology of HIV, transgenic mice expressing the CD4 receptor and CCR5 coreceptor have been developed. Despite being receptive to HIV infection, murine monocytes did not allow viral replication, indicating that additional human cellular components are required for successful infection. Consequently, the pathogenesis may not progress as it does in humans, even when mice express the corresponding viral entry receptor [114].

Nevertheless, the combination of transgenic and hu-PBL platforms to research hepatotropic viruses remains challenging, such as with HBV. While transient expression human receptors in the murine liver becomes more difficult, constitutive expression may be possible [115].

Hu-PBL models have the advantage of faithfully mimicking the so-called patient memory immune response in the mouse. In particular, it was demonstrated that the hu-PBL/DC model enables the investigation of correlates of immunity in EVD survivors [43], which might be expanded to the investigation of other viral infections, such as infection with Rift Valley fever virus (RVFV) [116], SARS-CoV-2 infection and reinfection [117,118,119,120,121,122,123,124,125,126], or yellow fever virus (YFV) infection [127]. In this way, it is also possible to analyze the interactions that take place between the immune cells throughout the development of a specific disease given that they have been demonstrated to enable the sequential engraftment of distinct cell types [39].

These models also seem to be appropriate for studying the effectiveness of new vaccination strategies [128,129], some of which are currently in the clinical stage. This will enable researchers to look at how vaccines affect the maturation of T cells and other immune system compartments in addition to the antibody response. The results coming from these investigations will help to improve therapy design, including the use of dendritic cells as vaccines, which has been shown to be effective against several viruses [69,130,131,132,133,134,135,136,137,138,139,140,141].

A third interesting approach that could be developed in PBL models is the studying of viral coinfections, which can arise in humans as a result of acute or chronic infections, such as those caused by HIV and other viruses [142,143,144], CMV and HBV [145], or SARS-CoV-2 and influenza viruses [146,147]. It would be attractive to learn how a past infection may influence the outcome of a subsequent one with a different virus, or vice versa, to develop better therapeutical approaches for those patients.

Finally, humanized models based on human liver cell transplantation (hu-liver) have been created to research viruses indicating liver tropism, such as HCV and HBV. These platforms have produced intriguing discoveries but lack the immunological component needed to fully understand the diseases that they cause [148,149,150]. Human hepatocytes and HSC have been transplanted into different mouse models to rebuild the immune system [151,152,153], but this has been only performed in HSC models, which lack the immune memory of the donor cells. Considering this, integrating the hu-PBL and hu-liver models would offer a potentially valuable tool for understanding viral chronic hepatitis and associated disorders, such as hepatocellular carcinoma.

## Figures and Tables

**Figure 1 viruses-15-00228-f001:**
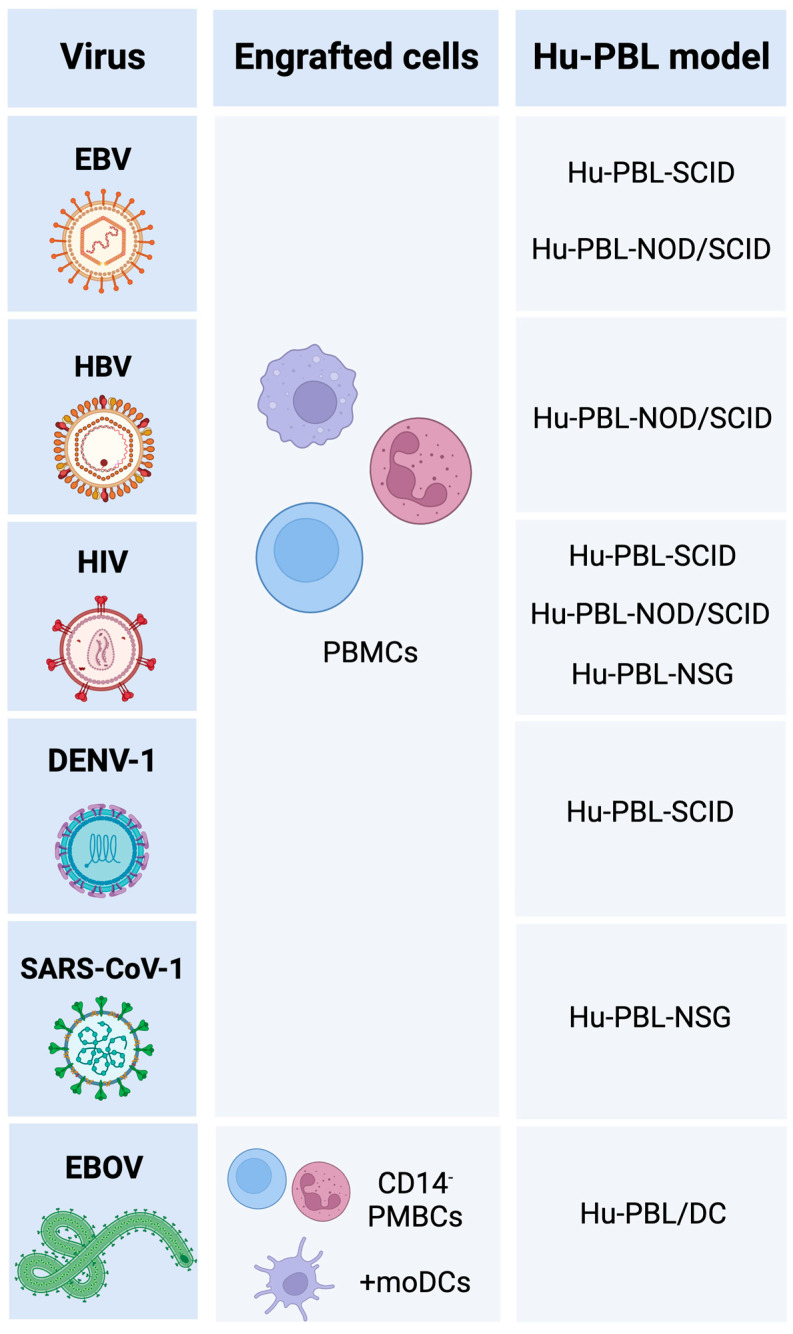
Hu-PBL mouse models used in research on the immunology of viral infections [8,22,24,37,38,43,44,45,46,47,48,49,50,51,52,53,54,55,56,57,58,59,60,61,62,63,64,65,66,67,68,69,70,71,72,73,74,75,76,77,78,79,80,81,82,83]. Created with BioRender.com.

## Data Availability

Not applicable.

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
