# Peer review of "Use of Hu-PBL Mice to Study Pathogenesis of Human-Restricted Viruses"

_viruses, 2023, doi:10.3390/v15010228_

Round 1
Reviewer 1 Report
Hu-PBL mouse models
- The authors should also add the original references on the premier research describing the development of the SCID mice (e.g., Bosma GC et al. Nature 1983 and Bosma MJ, Carroll AM. Annu Rev Immunol 1991)
- The authors may wish to include anti-asialo-ganglioside 1 as an additional method which is widely used to deplete NK cells and improve human cells engraftment in the models (anti-asialo-ganglioside 1 references: Kasai, M., et al. Nature 1981; Habu, S. et al. J. Immunol 1981 and Baiocchi, R., et al. JCI 2001)
- In lines 135-136, reference 32, could the authors confirm if mice in this study were engrafted with CD14-depleted PBMCs or with total PBMCs?
Research in virus immunology using Hu-PBL mouse models
- In lines 162-165, the authors nicely discussed the effect of cytokines (IL2 and IL10) and antibodies (anti-CD40 and anti-CD20) on lymphoma development in the models. It will strengthen the manuscript if the authors provide more detail on these studies and reference the original research papers and not only the review article (reference 36).
Conclusions and Perspectives
- It will strengthen the manuscript if the authors include additional limitations to the human virus’s infection of the mice in these models and how it deviated from the national host, as only a few are currently mentioned. E.g., route of infection and disease progression.
Comment on the sections numbering: Conclusions and Perspectives section should be number 4. The previous sections were numbered as follow:
- Instruction
- Hu-PBL mouse models
- Research in virus immunology using the Hu-PBL mouse model
There are no other sections with numbers 4 or 5.
Reviewer 2 Report
In this manuscript the authors review different ‘humanized’ mouse models, and how they have been applied to virology research. In my opinion, the topic is very interesting, and the revision and analysis of the data is appropriate, with an extensive and rigorous revision of the literature.
However, I found some minor points to clarify:
1) In the introduction section, the authors stated, “in particular, mouse models are frequently used since they are easy to handle cheap to maintain, and there are”. I think that a semicolon is missing in this line.
2) In the introduction section there are no references to related and previous work.
3) Line 57, “Despite that transplanted PBMCs contain also hematopoietic stem cells (HSC), model, minimal immune system”. I do not understand the word “model” in this line.
4) Line 197 “many drug-based therapeutic options to maintain a constant negative viral levels”. I think it should be “level”.
5) Line 371, “virus replication did not proceed in most of the cases, since DENV-1 was isolated once from one sample taken from one of the four infected mice” I do not understand this sentence
6) Line 434, “the main disadvantage is this the fact that” I do not understand the words “is this”.
